# Enhancement of Light Extraction Efficiency for InGaN/GaN Light-Emitting Diodes Using Silver Nanoparticle Embedded ZnO Thin Films

**DOI:** 10.3390/mi10040239

**Published:** 2019-04-10

**Authors:** Po-Hsun Lei, Chyi-Da Yang, Po-Chun Huang, Sheng-Jhan Yeh

**Affiliations:** 1Institute of Electro-Optical and Material Science, National Formosa University, No. 64, Wunhua Rd., Huwei, Yunlin County 632, Taiwan; d917205@oz.nthu.edu.tw (P.-C.H.); gy31999@gmail.com (S.-J.Y.); 2Department of Microelectronics Engineering, National Kaohsiung University of Science and Technology, Kaohsiung 811, Taiwan; cdyang@stu.nkmu.edu.tw

**Keywords:** Liquid phase deposition method, InGaN/GaN light-emitting diode, silver nanoparticle, zinc oxide, localized surface plasmon

## Abstract

In this study, we propose a liquid-phase-deposited silver nanoparticle embedded ZnO (LPD-Ag NP/ZnO) thin film at room temperature to improve the light extraction efficiency (LEE) for InGaN/GaN light-emitting diodes (LEDs). The treatment solution for the deposition of the LPD-Ag/NP ZnO thin film comprised a ZnO-powder-saturated HCl and a silver nitrate (AgNO_3_) aqueous solution. The enhanced LEE of an InGaN/GaN LED with the LPD-Ag NP/ZnO window layer can be attributed to the surface texture and localized surface plasmon (LSP) coupling effect. The surface texture of the LPD-Ag/NP ZnO window layer relies on the AgNO_3_ concentration, which decides the root-mean-square (RMS) roughness of the thin film. The LSP resonance or extinction wavelength also depends on the concentration of AgNO_3_, which determines the Ag NP size and content of Ag atoms in the LPD-Ag NP/ZnO thin film. The AgNO_3_ concentration for the optimal LEE of an InGaN/GaN LED with an LPD-Ag NP/ZnO window layer occurs at 0.05 M, which demonstrates an increased light output intensity that is approximately 1.52 times that of a conventional InGaN/GaN LED under a 20-mA driving current.

## 1. Introduction

Because gallium-nitride (GaN)-based blue light-emitting diodes (LEDs) have the inherent advantages of a wide bandgap in the green to ultraviolet range, a relatively long lifetime, and low energy consumption, they have received intensive and extensive investigation. InGaN/GaN LEDs with a high light output intensity and low power consumption have been widely applied to products such as solid-state lighting, backlight units for liquid crystal displays, car headlights, traffic lights, and full-color displays [1,2,3,4]. More recently, GaN-based LEDs and solid-state lighting were widely used as transmitting devices in the visible light communication (VLC) system, which has the characteristics of high-speed light communication with low transmission loss, the absence of electromagnetic interference, and license and high security [5,6,7]. A key issue for GaN-based LEDs and solid-state lighting in these applications is a high external quantum efficiency. The external quantum efficiency (EQE), which is the product of the internal quantum efficiency (IQE) and light extraction efficiency (LEE), determines the light output intensity of InGaN/GaN LEDs. The IQE is affected by the concentration of defects, amount of overflow carriers, and spreading area of injection current in the active region. To obtain a modern InGaN/GaN LED with a high IQE, various device structures, such as a designed active region including double-heterostructures and multiple quantum wells (MQWs), the insertion of an electron-blocking layer, and an expanded current-spreading layer, have been investigated [2]. To date, IQEs of higher than 80% have been achieved for InGaN/GaN LEDs because of the rapid development of growth methods and technologies [8,9,10]. The LEE of InGaN/GaN LEDs is also a crucial factor in determining the EQE. The LEE for InGaN/GaN LEDs is very low because of the high index contrast between GaN (n = 2.5) and free space. Several studies have reported that the IQE and LEE of InGaN/GaN LEDs can be enhanced using a patterned sapphire substrate to reflect and adjust the trajectory of light emitting to the sapphire substrate [11,12,13] or flip-chip structured-LEDs to increase the probability of light escaping from the sapphire [14,15]. However, the mechanically and chemically strong nature of sapphire makes patterning a challenging task. In addition, achieving the small dimensions of scattering objects through photolithography technologies is impossible because of the short wavelength of nitride-based LEDs. Some studies [16,17,18,19,20,21] have proposed the fabrication of textured GaN or a window layer to improve the LEE of InGaN/GaN LEDs through the changed trajectory of incident light beyond the escape cone. However, the textured InGaN/GaN LEDs demonstrate deteriorated electrical and temperature properties because of the inhomogeneous spatial field distribution under driving. Moreover, controlling the process and achieving a high repeatability of the rough window layer are difficult tasks. Some studies have reported that photonic crystals [22] and nanopyramids [23] can increase the LEE of InGaN/GaN LEDs through either diffraction or resonant coupling. However, these approaches are not practical because they require technologies that can accurately control the dimension, such as e-beam lithography, which is not suitable for obtaining a high throughput.

Recently, the surface plasmon (SP) coupling effect, which can improve the extraction efficiency of InGaN/GaN LEDs, has received considerable research attention [24,25,26]. SP coupling effects are the collective oscillations of electrons at the interface of a metal and a dielectric, and they can be classified as surface plasmon polaritons (SPPs) at metal surfaces and localized surface plasmons (LSPs) of local oscillation among isolated metallic nanostructures (NPs) with a resonant frequency. The SPP-enhanced LEE of InGaN/GaN LED can be realized by placing a metal layer near InGaN/GaN MQWs [27,28]. However, the inherently high reflection of the metal may block light emission from the InGaN/GaN active region, thus degrading the light output intensity. In contrast, several reports [23,24] have indicated that the LEE of InGaN/GaN blue LEDs can be improved by LSPs. Because the emission wavelength of InGaN/GaN blue LEDs matches the resonance wavelength of LSPs, the photons emitted from the InGaN/GaN active region can be closely coupled with the LSP modes to enhance the light output intensity of InGaN/GaN LED [29,30,31]. The resonant wavelength for the LSPs depends on the metal, such as silver (Ag) for blue, gold (Au) for green, and aluminum (Al) for UV [32,33]. In addition, because the interface is sufficiently rough to scatter the SPs, the coupling energy can be transferred into free space photons.

A well-known method to obtain Ag NPs is the solid-state dewetting process, which transforms an Ag thin film into Ag NPs under a high annealing temperature. However, a high-temperature process may result in the redistribution of the dopant and thermal stress, degrading the performance of devices. We are the first to synthesize a liquid-phase-deposited Ag nanoparticle embedded zinc oxide (LPD-Ag NP/ZnO) thin film through a chemical reaction in an aqueous solution at a low deposition temperature (i.e., even at room temperature) [34]. This method offers several advantages, including a low growth temperature, low cost, large area growth, good step coverage, and simple process and deposition equipment. In this study, the textured surface and LSP coupling effect, which result from the Ag NPs and ZnO in LPD-Ag NP/ZnO thin films, can enhance the LEE of InGaN/GaN LEDs with the LPD-Ag NP/ZnO window layer. The surface texture and LSP coupling effect of an LPD-Ag NP/ZnO thin film depend on the concentration of the silver nitrate (AgNO_3_) aqueous solution, which determines the root-mean-square (RMS) roughness, Ag content, and Ag NP size of the LPD-Ag NP/ZnO thin film. The measured results indicate that the main factor for enhancing the LEE of InGaN/GaN LEDs with the LPD-Ag NP/ZnO window layer is the LSP coupling effect.

## 2. Materials and Methods 

### 2.1. Fabrication of InGaN/GaN LEDs with LPD-Ag NPs/ZnO Thin Films

GaN epi-wafers were grown on a c-face (0001) sapphire substrate using a metal-organic chemical vapor deposition (MOCVD) system [29]. The InGaN/GaN LED structure comprises a GaN buffer layer grown at a low temperature, a heavily Si-doped n-type GaN layer, an InGaN/GaN MQW active region, and an Mg-doped p-type GaN layer. Indium-tin-oxide (ITO; Tyntek Co., Hsinchu, Taiwan) was deposited on the p-type GaN layer to form a transparent conductive layer (TCL). The epi-wafers were then patterned using standard photolithographic and partial etching to define the emitting regions. A Ti/Pt/Au alloy was used as ohmic contact metal in the p- and n-GaN contact regions, and the wafer was alloyed in an N_2_ atmosphere for 5 min at 450 °C. The finished wafer was placed in the treatment solution to deposit the LPD-Ag NPs/ZnO window layer. The size of the emission window for the InGaN/GaN LEDs with ITO TCL and an LPD-Ag NPs/ZnO window layer was 300 × 300 μm^2^. The schematics of the InGaN/GaN LED with an LPD-Ag NP/ZnO window layer are shown in Figure 1.

### 2.2. Preparation of Treatment Solution and Growth of LPD-Ag NP/ZnO Thin Films

The treatment solution for LPD-Ag NP/ZnO thin films grown on the sapphire substrate was prepared according to the following steps. First, ZnO powder (J. T. Baker) was added into deionized water-diluted HCl (12 M, Taimax, Taiwan), and the mixed solution was then stirred for 1 h at 25 °C to ensure that the HCl solution was saturated with the ZnO powder. Second, a aqueous solution of AgNO_3_ was added to the ZnO-saturated HCl solution to form the treatment solution. The pH scale of the treatment solution was dominated by the concentration of HCl. The optimum pH scale to grow the LPD-Ag NP/ZnO thin film on the sapphire substrate was approximately 4–5.5. The LPD system for the deposition of the LPD-Ag NP/ZnO thin films comprised a Teflon vessel immersed in a temperature-controlled water bath system. The temperature-controlled water bath system can provide a uniform temperature distribution with an accuracy of 0.1 °C. The following process was implemented to deposit the LPD-Ag NP/ZnO thin film on the InGaN/GaN LED. First, a Teflon vessel containing the treatment solution was placed in the temperature-controlled water bath system at 25 °C for 3–5 min. This preheating step ensured a uniform deposition of the LPD-Ag NP/ZnO thin film on the inserted substrate. Second, the InGaN/GaN LED was inserted in the treatment solution for LPD-Ag NP/ZnO thin film deposition. The surface morphology of LPD-Ag NPs/ZnO on the sapphire substrate was investigated through scanning electron microscopy (SEM), and the Ag content in the LPD-Ag NP/ZnO thin film was detected through energy-dispersive X-ray spectroscopy (EDS).

## 3. Results and Discussion

To obtain a fair deposition rate, which determines the RMS roughness of the LPD-ZnO thin film, the pH scale of treatment solution and diluted HCl concentration should be maintained at 4–5.5 and 6 M, respectively, during the deposition process and the deposition temperature needs a temperature distribution with an accuracy of 0.1 °C. Figure 2 shows the SEM images for the LPD-ZnO thin film (Figure 2a) and those for the LPD-Ag NP/ZnO thin films deposited at AgNO_3_ concentrations of 0.01, 0.05, and 0.1 M (Figure 2b–d). The estimated percentages of Ag NP size in the LPD-Ag NP/ZnO thin films deposited at AgNO_3_ concentrations of 0.05 and 0.1 M are shown in Figure 2e,f. The deposition temperature of the all thin films was 25 °C. The LPD-ZnO thin film exhibited a hexagonal and flake-shaped structure on the sapphire substrate, as shown in Figure 2a. Figure 2b–d show that the structure of the LPD-Ag NP/ZnO thin films reflected the randomly distributed Ag NPs on the hexagonal and flake-shaped ZnO. In addition, Figure 2b–d show that the Ag NP distribution of the LPD-Ag NP/ZnO thin film deposited at a high AgNO_3_ concentration is denser than that of the thin film deposited at a low AgNO_3_ concentration. The average size of the randomly distributed Ag NP on the flake-shaped ZnO was approximately 76.8 and 142.6 nm for AgNO_3_ concentrations of 0.05 and 0.1 M, respectively. The chemical reaction for the LPD-deposited ZnO thin film can be expressed as follows [34,35]:2ZnO + HCl + 2H_2_O ⇆ Zn(OH)_4_^2−^ + Zn^2+^ + H^+^ + Cl^−^,(1)
Zn(OH)_4_^2^ ⇆ ZnO_2_^2−^ + H_2_O,(2)

The concentration of zinc species including Zn^2+^ and Zn(OH)_4_^2−^ was used to determine the growth rate of the LPD-ZnO thin film in our previous study [35]. AgNO_3_ powder can dissolve in H_2_O and form an Ag-complex, namely Ag(OH)_2_^−^, in AgNO_3_ aqueous solution through the following chemical reactions [36]:AgNO_3_ ⇆ Ag^+^ + NO_3_^−^,(3)
Ag^+^ + 2H_2_O ⇆ Ag(OH)_2_^−^ + 2H^+^,(4)

The Ag-complex reacts with Zn^2+^ and ZnO_2_^2−^ to form the LPD-Ag NP/ZnO thin film according to the following equation [35,36]:ZnO_2_^2−^ + Zn^2+^ + 2Ag(OH)_2_^−^ ⇆ 2ZnO + 2Ag + 2H_2_O + 2OH^−^,(5)

The formation of a flake-shaped ZnO thin film depends on the pH scale of the treatment solution [31]. To obtain a similar flake-shaped ZnO thin film, the pH scale of the treatment solution for LPD-Ag NP/ZnO thin films grown on the sapphire substrate was maintained at 4–5.5. The concentration of Ag^+^ and Ag(OH)_2_^−^ in treatment solution depends on the concentration of AgNO_3_. A high AgNO_3_ concentration can increase the concentration of Ag(OH)_2_^−^, resulting in large and dense Ag NPs embedded on the flake-shaped ZnO.

Table 1 lists the Ag contents of LPD-Ag NP/ZnO thin films measured using EDS for various AgNO_3_ concentrations. The Ag content of LPD-Ag NP/ZnO thin films increased with the AgNO_3_ concentration, resulting from a high Ag^+^ concentration, as shown in Equation (3). Moreover, the density of Ag NPs distributed on the flake-shaped ZnO for LPD-Ag NP/ZnO thin films observed from the SEM images exhibited a similar trend to the Ag content of the LPD-Ag NP/ZnO thin films shown in Table 1. We quantitatively determined the density of the Ag NP distributed on the flake-shaped ZnO according to the Ag content of the LPD-Ag NP/ZnO thin films, and we used the varying AgNO_3_ concentrations of the treatment solution to obtain the specific distribution and size of Ag NPs on the flake-shaped ZnO. However, adding AgNO_3_ to the treatment solution might cause Ag atoms to dope into the flake-shaped ZnO, rather than form an Ag NP embedded on the flake-shaped ZnO during LPD-Ag NP/ZnO thin film deposition. The extinction-peak position depends on the Ag NP diameter and distance between Ag NPs [32]. Increasing the diameter of Ag NP or decreasing the distance between Ag NPs will shift the extinction-peak wavelength to a long value. Therefore, Ag NPs with a large size or high content under a high AgNO_3_ concentration result in a red-shift for the extinction-peak, as shown in Table 1. The insets of Figure 2a,c show the EDS spectra for the LPD-ZnO and LPD-Ag NPs/ZnO with an AgNO_3_ concentration of 0.05 M. The O and Zn atoms contents of the LPD-ZnO thin film were 72.77 and 27.23 at %, respectively, and the compositions of O, Zn, and Ag atoms in the LPD-Ag NPs/ZnO were 70.59, 25.64, and 3.76 at %, respectively. The ratio of O to Zn for the LPD-ZnO thin film (2.67) was similar to that in the LPD-Ag NPs/ZnO thin film (2.75), implying that the Ag NPs were randomly distributed on the flake-shaped ZnO rather than doped during the deposition process.

The photons emitted from the InGaN/GaN active region to the LPD-Ag NP/ZnO window layer were localized along the interface of the randomly-distributed Ag NPs and ZnO. The near-field strength at this interface was enhanced and the incident light from the InGaN/GaN active region was coupled with the LSPs in the Ag NPs. The enhanced light coupled with the LSPs was then decoupled by the Ag NPs and re-emitted into the free space [34]. The wavelength of incident light from the InGaN/GaN active region should be close to the LSP resonant wavelength to enhance the LEE. To determine the LSP resonance wavelength related to the Ag NPs, the measured extinction spectra of the LPD-Ag NP/ZnO thin film are shown in Figure 3 as functions of the AgNO_3_ concentration. In addition, the diameter of the Ag NPs, the Ag content of the LPD-Ag NP/ZnO thin film, and the peak position of the aforementioned extinction spectra are summarized in Table 1. The peak position of the extinction spectra in Figure 3 reveals an obvious red-shift as the AgNO_3_ concentration increases from 0.03 to 0.1 M. This is attributed to the following reasons. First, the resonant electromagnetic behavior of the Ag NPs depends on the confinement of conduction electrons to the small particle volume [37]. Because the extinction of light with the wavelength is smaller than that of the diameter of Ag NPs, the electrons move and resonate at a specific frequency (or wavelength), which is called the particle dipole plasmon frequency (or wavelength). As the size of the Ag NPs increases, the conduction electron cannot move in phase, leading to a reduced depolarization field for the red-shift phenomenon. The size of the Ag NPs increases with increasing AgNO_3_ concentration, resulting in a reduced depolarization field, as indicated in Table 1. Second, the peak position of the absorption or extinction spectrum for Ag NP/ZnO thin films, which depends on the LSP resonant wavelength, can be represented as follows [38,39]:(6)$$\lambda_{P}=\sqrt{\frac{4\pi^{2}c^{2}m_{\mathrm{eff}}\varepsilon_{0}}{{\mathrm{Ne}}^{2}}},$$
where λ_P_ is the LSP-resonant wavelength, m_eff_ is the effective mass of the free electron of the metal, and N is the electron density of metal. The LSP resonant wavelength relates to the electron density of metal. The Ag content increases from 1.64 to 9.46 at % as the AgNO_3_ concentration increases from 0.03 to 0.1 M, as shown in Table 1. The electron density of the Ag NPs decreases with increasing Ag content (AgNO_3_ concentration) because electrons move from the Ag NPs to the conduction band of the flake-shaped ZnO, as shown in the inset of Figure 3. According to Equation (6), the reduced electron density of Ag NPs leads to a red-shift phenomenon [40].

To obtain the optimal AgNO_3_ concentration, which determines the Ag NP size, to maximize LEE, we used the OptiFDTD computational software, which is based on the finite-difference time-domain, to calculate the optical intensity for InGaN/GaN LEDs with or without the Ag NP/ZnO window layer. A random distribution of Ag NPs with various diameters on the flake-shaped ZnO window layer in InGaN/GaN LED was applied to determine the Ag NP distribution in the LPD-Ag NP/ZnO thin film, as shown in Figure 2. Figure 4a shows the calculated optical intensity of conventional InGaN/GaN LEDs, InGaN/GaN LEDs with the flake-shaped ZnO window layer, and InGaN/GaN LEDs with Ag NPs distributed on the flake-shaped ZnO window layer for Ag NPs with various diameters. The emitting wavelength was set at 460 nm. Figure 4 indicates that the InGaN/GaN LED with Ag NPs distributed on the flake-shaped ZnO thin film achieves the maximum optical intensity when the diameter of the Ag NPs is 75 nm. Figure 4b,c show the electric field distribution of InGaN/GaN LEDs with LPD-ZnO and 75-nm-sized LPD-Ag NP/ZnO window layers. Although the color distribution in Figure 4c is similar to Figure 4b, the scale of the color bar in Figure 4c is larger than that in Figure 4b. These results indicate that an LSP coupling effect is found in InGaN/GaN LED with an LPD-Ag NP/ZnO window layer. Consequently, the enhanced light intensity or LEE might be attributed to the LSP coupling effect of the Ag NPs and the textured LPD-Ag NPs/ZnO thin films, which increase the probability of photons escaping from p-GaN.

The current–voltage (I–V) and light output intensity–current (L–I) characteristics for the conventional, LPD-ZnO-enhanced, and LPD-Ag NP/ZnO-enhanced InGaN/GaN LEDs are shown in Figure 5a. The forward voltages for the InGaN/GaN LEDs with an LPD-ZnO window layer and LPD-Ag NP/ZnO window layers grown at AgNO_3_ concentrations of 0.01, 0.03, 0.05, 0.07, and 0.1 M were 3.64, 3.62, 3.65, 3.67, 3.66, and 3.66 V, respectively, at 20 mA, whereas that for the conventional InGaN/GaN LED was 3.58 V at the same driving current level. The forward voltage of the InGaN/GaN LEDs with LPD-ZnO and LPD-Ag NP/ZnO window layers was higher than that for the conventional InGaN/GaN LED because of the deterioration of the ohmic contact resulting from the LPD-ZnO and LPD-Ag NP/ZnO growth on the periphery of the contact. In addition, the series resistance of the InGaN/GaN LEDs with LPD-ZnO and LPD-Ag NP/ZnO window layers was slightly higher than that of conventional InGaN/GaN LED. The series resistance of the InGaN/GaN LEDs depends on the contact metal, metal/ITO interface, ITO, p-GaN, InGaN/GaN active region, and n-GaN, and the increased resistance is caused by the deterioration in the contact during LPD-Ag NP/ZnO deposition. The light output intensities for InGaN/GaN LEDs with an LPD-ZnO window layer and LPD-Ag NP/ZnO window layers grown at AgNO_3_ concentrations of 0.01, 0.03, 0.05, 0.07, and 0.1 M were 132.9, 133.3, 133.8, 143.5, 136.2, and 134.2 mcd, respectively, for a 20-mA driving current, whereas that for the conventional InGaN/GaN LED was 95.0 mcd at the same driving current. The insets of Figure 5a show the microscope images of light emission for InGaN/GaN LEDs with and without an LPD-Ag NP/ZnO window layer. The light output intensity of the InGaN/GaN LED with an LPD-Ag NP/ZnO window layer was higher than that without an LPD-Ag NP/ZnO window layer because of the improved LEE. Figure 5b represents the calculated light intensity (from Figure 4a) and measured light output intensity (from Figure 5a) under the driving current of 100 mA as a function of Ag NP size. Figure 5c performs the transmittance spectra of LPD-Ag NP/ZnO thin films as a function of AgNO_3_ concentration over the visible wavelength and the inset of Figure 5c shows the transmittance of LPD-ZnO and LPD-Ag NP/ZnO thin films with the changed AgNO_3_ concentration at the wavelength of 460 nm. The calculated and measured results indicate that the highest light intensity occurred at the Ag NP size of 75 and 76.8 nm (AgNO_3_ concentration of 0.05 M) because of the highest transmittance at the 460-nm wavelength shown in the inset of Figure 5c. The highest transmittance can possibly be attributed to the LSP effect [41]. However, the calculated light intensity decreases with Ag NP size increasing form 0 to 60 nm, but the measured light output intensity increases slightly at the same range. This may be attributed to the close transmittance for LPD-ZnO and LPD-Ag NP/ZnO with the Ag NP size of 60 nm, as shown in the inset of Figure 5c. A low Ag NP density on fake-shaped ZnO and a weak LSP effect lead to a slight increase in light output intensity for InGaN/GaN LED with the LPD-Ag NP/ZnO window layer (60-nm AgNP) compared with that with the LPD-ZnO window layer.

The increased light output intensity may be attributed to the textured surface and LSP coupling effect. It is important to confirm the dominated factor of enhanced LEE for InGaN/GaN LED with an LPD-Ag NP/ZnO window layer. Figure 6 represents the RMS roughness for LPD-Ag NP/ZnO thin films as a function of AgNO_3_ concentration. The insets of Figure 6 show the AFM images for LPD-Ag NP/ZnO thin films grown at the AgNO_3_ concentration of 0.01, 0.05, and 0.1 M. In our previous study [31], the LEE of InGaN/GaN LED with an LPD-ZnO window layer depended on the surface texture rather than the ratio of Zn/O. The surface texture depends on the deposition rate of ZnO flake, which was determined by the HCl concentration, and the ratio of Zn/O is almost the same under the varied HCl concentration. Therefore, the enhanced LEE of InGaN/GaN LED caused by the textured LPD-Ag NP/ZnO window layer can be investigated by the RMS roughness. The LEE of an LED gives the ratio of the number of useful emitting-photons to the number of injection charge particles and is defined as [3]
(7)$$\eta_{\mathrm{LEE}}=\frac{\mathrm{dP}}{\mathrm{dI}}\frac{e\lambda }{hc},$$
where P is the light output power, I is the injection current, h is the Plank constant, and λ is the emitting wavelength. A textured window layer applied to InGaN/GaN LED can improve the escaped probability of the photons emitted from the InGaN/GaN active region. The RMS roughness of the LPD-Ag NP/ZnO thin film decreases with an increase in the AgNO_3_ concentration. This is attributed to the following reasons. First, increasing the AgNO_3_ concentration leads to a large number of Ag NPs aggregated at the interface between ZnO flakes to reduce the RMS roughness of the LPD-Ag NP/ZnO thin film. Secondly, the ZnO flakes in LPD-Ag NP/ZnO thin films tend to bend under a high AgNO_3_ concentration, resulting in a low RMS roughness. This is attributed to the fact that the concentration of the Ag precursor has a great effect on the morphology of ZnO [42]. LED with a weak textured window layer (with low RMS roughness) shows a lower LEE than that with a strong one (with large RMS roughness) [3]. However, the light output intensity of InGaN/GaN LED increases as the AgNO_3_ concentration rises from 0 to 0.05 M, as shown in Figure 5. Consequently, the increasing light output intensity of InGaN/GaN LED with an LPD-Ag NP ZnO window layer is attributed to the LSP coupling effect, which occurred because the extinction wavelength was similar to the wavelength of the photon emitted from the InGaN/GaN active region, as shown in Figure 3 [41]. However, further increasing the AgNO_3_ concentration above 0.07 M reduced the light output intensity because of the weak LSP coupling effect caused by the difference between the resonant wavelength and the emission wavelength from the InGaN/GaN active region. In addition, a strong LSP coupling effect can be obtained by choosing an extinction wavelength similar to and smaller than the emission wavelength from the LED active region. 

Figure 7 shows the average forward voltage and light output intensity under an injection current of 20 mA for the chosen chips from the InGaN/GaN wafer with the optimal LPD-Ag NP/ZnO and LPD-ZnO window layer for varied runs. The uniformity of the LPD-Ag NP/ZnO thin film on the InGaN/GaN wafer is noteworthy because it is crucial when determining the performance of the InGaN/GaN LEDs. Because the size and distribution of LPD-Ag NPs on the flake-shaped ZnO were similar, the chosen chips from the InGaN/GaN wafer with the optimal LPD-Ag NP/ZnO window layer revealed variations of approximately 5% in the forward voltage and 4.3% in the light output intensity under the same driving current conditions. In addition, the standard deviation of the measured enhancement of emission intensity of the InGaN/GaN wafer with LPD-Ag NP/ZnO to that with LPD-ZnO was about 1.4%. 

Figure 8 shows the electroluminescence (EL) spectra of the InGaN/GaN LEDs with and without an LPD-Ag NP/ZnO window layer deposited at various AgNO_3_ concentrations under a 20-mA driving current. The LPD-Ag NP/ZnO thin film deposited at an AgNO_3_ concentration of 0.05 M exhibited the highest light output intensity because of the LSP coupling effect, and it was approximately 1.52 times higher than that of the conventional InGaN/GaN LEDs. The inset of Figure 8 depicts the full width at half maximum (FWHM) of the EL spectra for the LPD-Ag NP/ZnO-enhanced and conventional InGN/GaN LEDs. The FWHM of the InGaN/GaN LED with an LPD-Ag NP ZnO window layer is narrower than that of a conventional InGaN/GaN LED because of the LSP coupling effect, which can concentrate the emitting spectrum at the resonant wavelength.

## 4. Conclusions

An LPD-Ag NP/ZnO window layer can enhance the light output intensity of InGaN/GaN LEDs. The improved LEE was attributed to the LSP coupling effect, which was caused by the Ag NPs distributed on the flake-shaped ZnO. The size and distribution of the Ag NPs on the flake-shaped ZnO were adjusted according to the concentration of the AgNO_3_ aqueous solution that was added to the ZnO-powder-saturated HCl aqueous solution and the pH scale of treatment solution. We quantitatively demonstrated the density of the Ag NPs distributed on the flake-shaped ZnO by determining the Ag content of the LPD-Ag NP/ZnO thin films, because they had similar variations in AgNO_3_ concentration. The experimental result indicated that the maximum light output intensity occurred when Ag NP with the diameter of approximately 76.8 nm, which is close to the calculated result of Ag NP with the diameter of 75 nm, and with the content of 3.76 at %, were distributed on the flake-shaped ZnO, because the LSP resonance wavelength for this condition was similar to the emission wavelength of the InGaN/GaN LEDs. The light output intensity of the InGaN/GaN LEDs with the optimal LPD-Ag NP/ZnO window layer was 1.52 times higher than that of the conventional InGaN/GaN LED.

## Figures and Tables

**Figure 1 micromachines-10-00239-f001:**
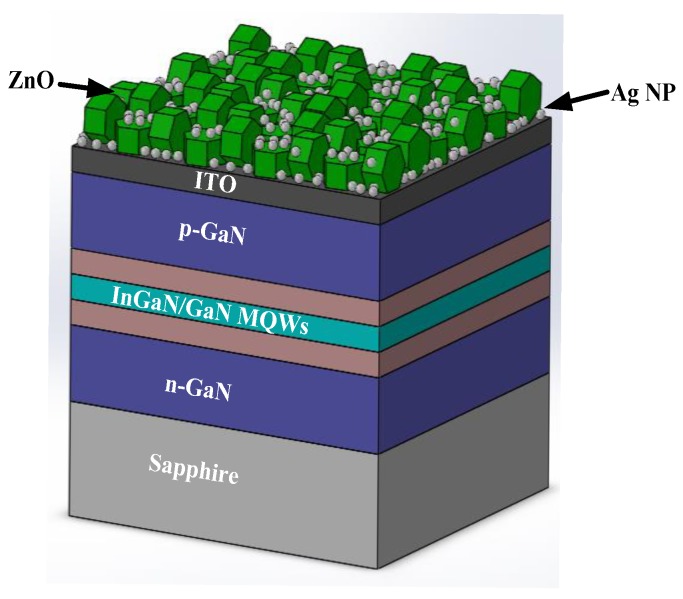
The schematics of InGaN/GaN LED with an LPD-Ag NP/ZnO window layer.

**Figure 2 micromachines-10-00239-f002:**
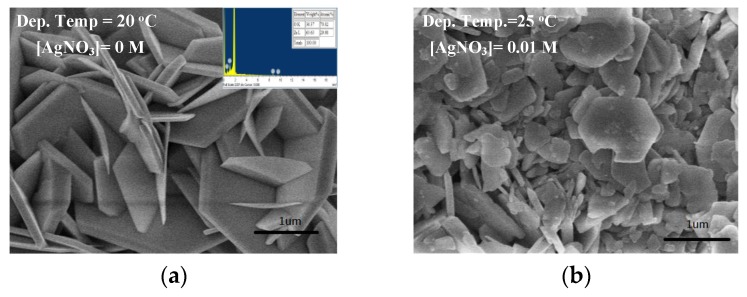
SEM images for (**a**) LPD-ZnO and LPD-Ag NP/ZnO thin films under AgNO_3_ concentrations of (**b**) 0.01, (**c)** 0.05, and (**d**) 0.1 M. The insets of (**a**) show the content of Zn and O of LPD-ZnO and (**c**) represent the content of Zn, O, and Ag and Ag NP size of LPD-Ag NP/ZnO under an AgNO_3_ concentration of 0.05 M. The estimated percentage of Ag NP size in LPD-Ag NP/ZnO thin films deposited at (**e**) 0.05 and (**f**) 0.1 M

**Figure 3 micromachines-10-00239-f003:**
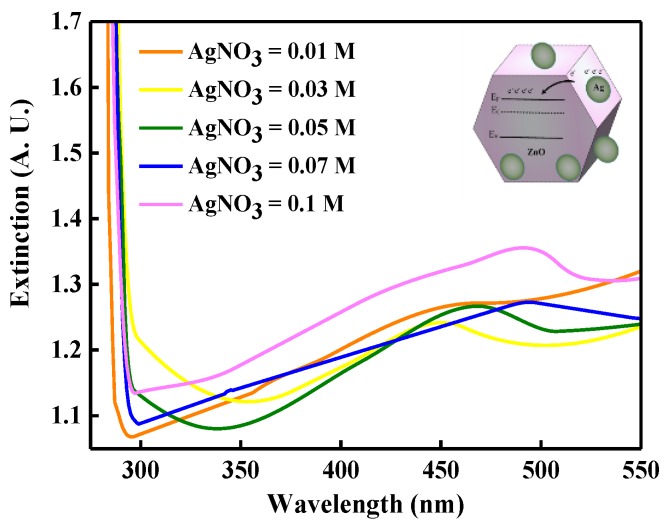
Measured extinction spectra of the LPD-Ag NP/ZnO thin films for different AgNO_3_ concentrations. The inset shows a schematic of the movement of electrons between Ag and ZnO.

**Figure 4 micromachines-10-00239-f004:**
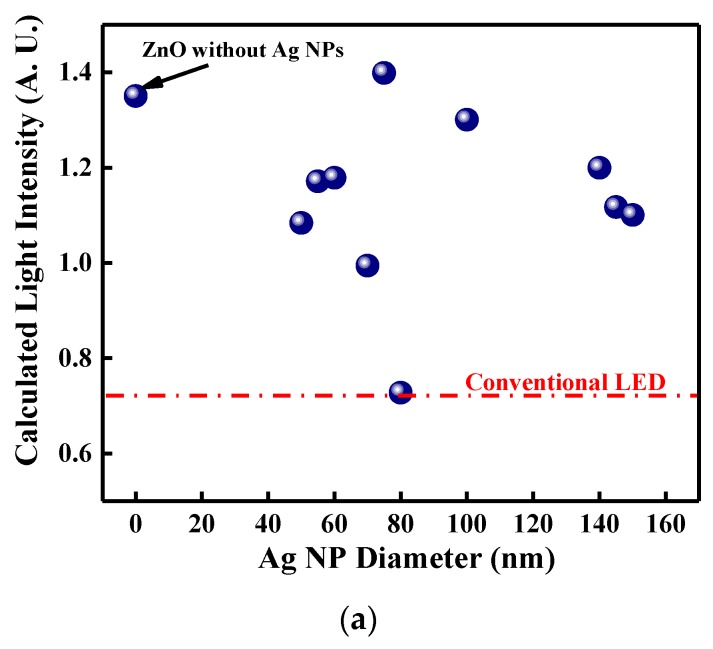
(**a**) Calculated optical intensity for the conventional InGaN/GaN LEDs, InGaN/GaN LEDs with flake-shaped-ZnO window layers, and InGaN/GaN LEDs with Ag NP/ZnO window layers for Ag NPs with various diameters. The electric field distribution of InGaN/GaN LEDs with (**b**) flake-shaped-ZnO window layers, and with (**c**) 75-nm-sized Ag NP/ZnO window layers.

**Figure 5 micromachines-10-00239-f005:**
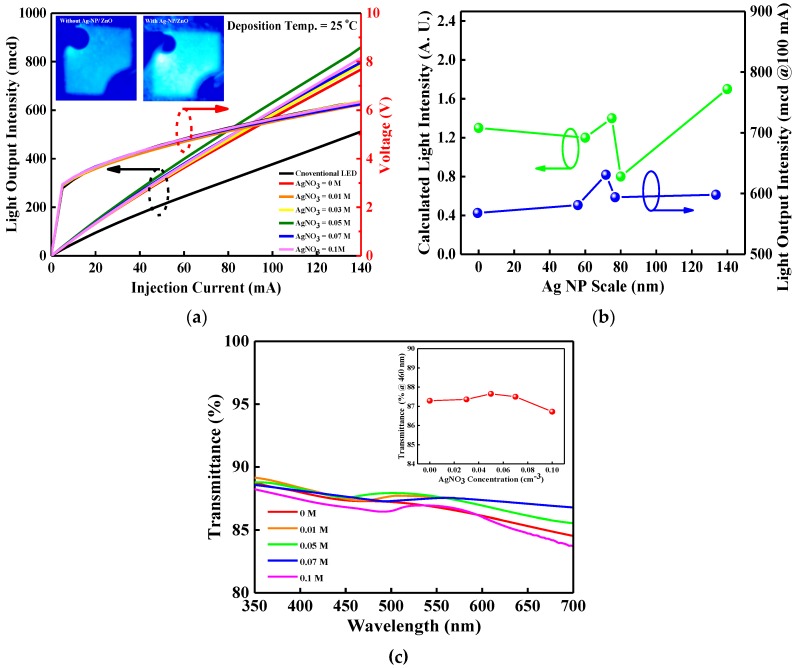
(**a**) I–V and L–I plots for the conventional, LPD-ZnO-enhanced, and LPD-Ag NP/ZnO-enhanced InGaN/GaN LEDs. The micrographs of light emission for InGaN/GaN LEDs with and without the LPD-Ag NP/ZnO window layer are shown in the inset, (**b**) calculated light intensity and measured light output intensity at a 100-mA driving current as a function of Ag NP size, and (**c**) transmittance spectra of LPD-Ag NP/ZnO thin films as a function of AgNO_3_ concentration over the visible range. The inset of Figure 5c shows the transmittance at a 460-nm wavelength for LPD-ZnO and LPD-Ag NP/ZnO thin films deposited at varied AgNO_3_ concentrations.

**Figure 6 micromachines-10-00239-f006:**
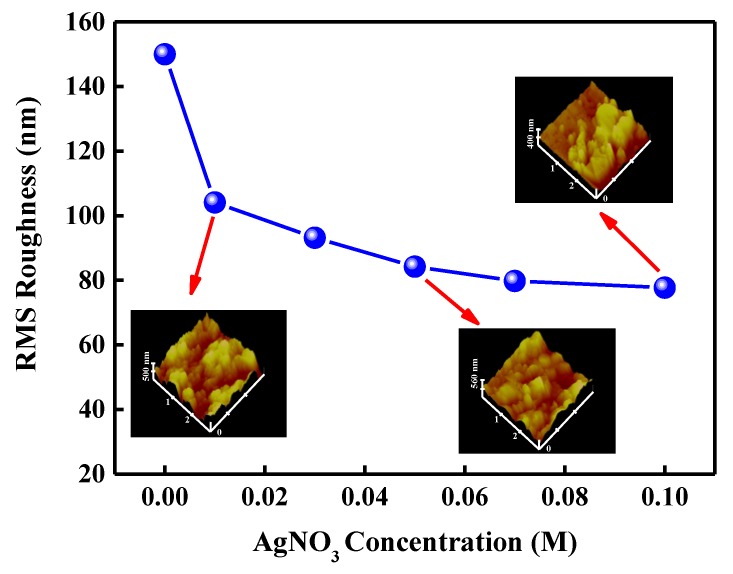
Root mean square (RMS) roughness for LPD-Ag NP/ZnO thin films as a function of AgNO_3_ concentration. The insets are AFM images for LPD-Ag NP/ZnO thin films grown at 0.01, 0.05, and 0.1 M.

**Figure 7 micromachines-10-00239-f007:**
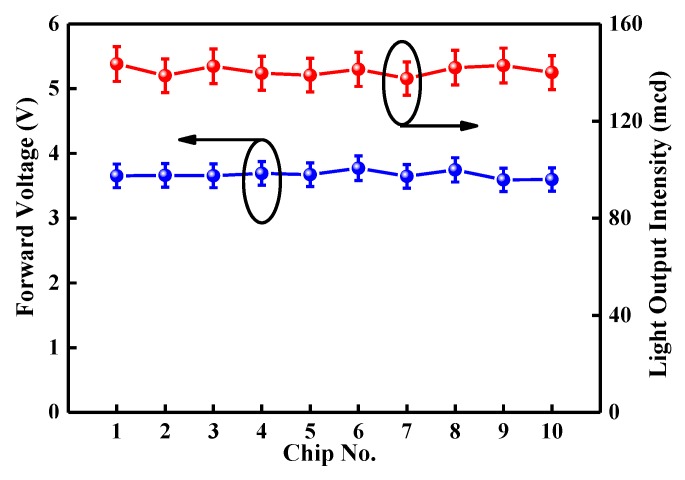
Average forward voltages and light output intensities at an injection current of 20 mA for the selected chips on the InGaN/GaN wafers with the optimal LPD-Ag NP/ZnO and LPD-ZnO window layer for varied runs.

**Figure 8 micromachines-10-00239-f008:**
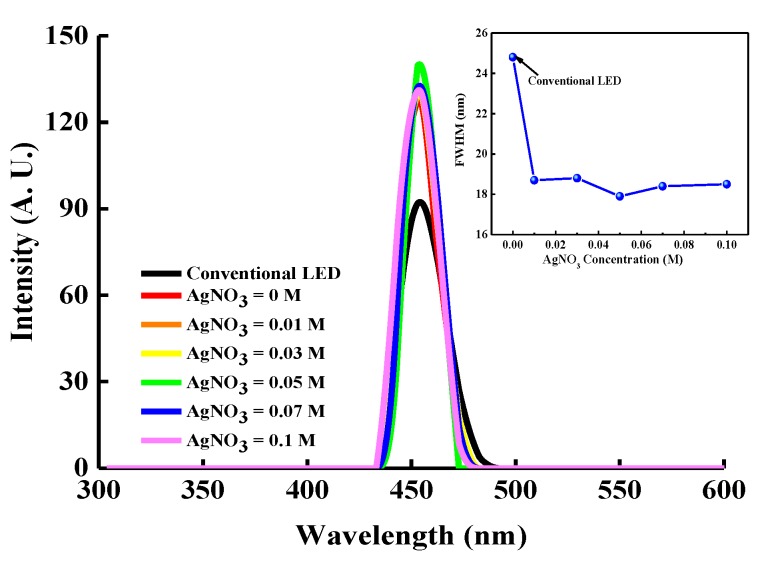
Electroluminescence (EL) spectra of InGaN/GaN LEDs with and without LPD-Ag NP/ZnO thin films deposited at different AgNO_3_ concentrations under a 20-mA driving current. The inset shows the FWHMs of the EL spectra.

**Table 1 micromachines-10-00239-t001:** Relationship between measured Ag NP diameter, Ag content, and extinction-peak position under different AgNO_3_ concentrations.

AgNO_3_ Concentration (M)	Measured Ag NP Diameter (Average) (nm)	Ag Content (at %)	Extinction-Peak Position (nm)
0.01	-	0.21	-
0.03	60.3	1.64	442
0.05	76.8	3.76	463
0.07	82.3	5.89	489
0.1	142.6	9.46	492

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
