# Peer review of "Enhancement of Light Extraction Efficiency for InGaN/GaN Light-Emitting Diodes Using Silver Nanoparticle Embedded ZnO Thin Films"

_micromachines, 2019, doi:10.3390/mi10040239_

Round 1

Reviewer 1 Report

1.       I would recommend authors to add the schematic diagram of the LSP-enhanced InGaN/GaN LED structure (side view) with metal nanoparticle embedded ZnO window layer. It makes paper more readable.

2.       According to EDX analysis flake-shaped ZnO films are non-stochiometric and oxygen-rich. This means that the interstitial oxygen and zinc vacancies are most probable intrinsic defects in ZnO. Did authors investigate the optical properties of ZnO, especially photoluminescence?  It is necessary to comment how does stoichiometric ratio in ZnO influence the light extraction efficiency.

Author Response

1. I would recommend authors to add the schematic diagram of the LSP-enhanced InGaN/GaN LED structure (side view) with metal nanoparticle embedded ZnO window layer. It makes paper more readable.

Rebuttal (R1): We have added the schematics of InGaN/GaN LED with LPD-Ag NP/ZnO window layer (Fig. 1) in page 3.

2. According to EDX analysis flake-shaped ZnO films are non-stochiometric and oxygen-rich. This means that the interstitial oxygen and zinc vacancies are most probable intrinsic defects in ZnO. Did authors investigate the optical properties of ZnO, especially photoluminescence?  It is necessary to comment how does stoichiometric ratio in ZnO influence the light extraction efficiency.

R2: In our previous study (ref. 31), the ratio of Zn/O for LPD-ZnO thin films deposited at varied HCl concentration is close. The LEE of InGaN/GaN LED with an LPD-ZnO window layer relates to the surface texture, which can be decided by the deposition rate and investigated by the RMS roughness. We have added this description in line 286-296 in page 9.

Reviewer 2 Report

In this manuscript, the authors investigated the solution-processed Ag-NP/ZnO layer for improving the light extraction efficiency of InGan/GaN LED. I have some comment on this manuscript.

  p.3. The size of fabricated Ag NPs can be estimated from the SEM image in Fig. 1 and the detailed discussion about the particle size.

p.5, 2nd paragraph. I cannot understand how to estimate the light extinction spectra in Fig. 2.

p.5, 2nd paragraph. Since the ZnO layer is not flat, the transmittance spectra of Ag-NP/ZnO should be added to discuss the detailed effect of SP.

p.6, Is the calculated light intensity in Fig. 3(a) all the wavelength ranges in Fig 2?

p.7. Please add the graph about the relationship between the LEE in Fig. 3(a) and the light output intensity in Fig. 4.

Author Response

1. p.3. The size of fabricated Ag NPs can be estimated from the SEM image in Fig. 1 and the detailed discussion about the particle size.

R1: We have added Fig. 2 (e) and (f) in page 5 to show the estimated percentage of Ag NP size in LPD-Ag NP/ZnO thin films deposited at AgNO3 concentration of 0.05 and 0.1 M

2. p.5, 2nd paragraph. I cannot understand how to estimate the light extinction spectra in Fig. 2.

R2: The light extinction spectra in Fig.3 (old Fig. 2) were measured results. We have added the "measured" in the caption and in line 194 in page 6.

3. p.5, 2nd paragraph. Since the ZnO layer is not flat, the transmittance spectra of Ag-NP/ZnO should be added to discuss the detailed effect of SP.

R3: We have added Fig. 5 (c) to show the transmittance spectra of LPD-ZnO and LPD-Ag NP/ZnO over the visible range and to discuss the SP effect in line 267-274  in page 8.

     4. p.6, Is the calculated light intensity in Fig. 3(a) all the wavelength ranges in Fig 2?

R4: The wavelength for calculated light intensity was set at 460 nm. We have added it in line 226 in page 7.

5. p.7. Please add the graph about the relationship between the LEE in Fig. 3(a) and the light output intensity in Fig. 4.

R5: We have added Fig. 5 (b) to show the relationship between calculated light intensity and measured light output intensity and discussion in line 260-274  in page 8.

Reviewer 3 Report

In this paper, the authors introduced a liquid-phase-deposited silver nanoparticle embedded ZnO  (LPD-Ag NP/ZnO) thin film at room temperature to improve the light extraction efficiency (LEE)  for InGaN/GaN light-emitting diodes (LEDs). The treatment solution for the deposition of LPD-Ag/NP ZnO thin film comprised a ZnO-powder-saturated HCl and a silver nitrate (AgNO3) aqueous solution. The enhanced LEE of an InGaN/GaN LED with the LPD-Ag NP/ZnO window layer can be attributed to the textured surface and localized surface plasmon (LSP) coupling effect. The idea behind this is interesting. However, I still have quite a number of concerns in this manuscript. There are times where there are not enough data to support the conclusions of the author. Please see some of the major concerns below.

1.       The information for the GaN LED structure is not enough. The authors should give much more information about this which means that authors need to add the full LED structure, so the readers can get its reproducibility. 

2.  The authors should give much more information about the novelty of this paper, especially the effect of using these materials as LED, which applications can be used this device?

3. The fabrication tolerance analysis, which can offer a good guide for the fabrication requirement, and the key parameters, need to be added in the results section.

4. More references need to be included in the introduction part to understand the applications of using GaN materials as a waveguide and LED as a light source in optical system.

1.      “Modeling a 1x8 MMI green light power splitter based on Gallium-Nitride slot waveguide structure", IEEE Photonics Technology Letters, 30(8), 2018 (720-723)

2.  "1x4 MMI visible light wavelength demultiplexer based on GaN slot waveguide structures"

     J. Photonics and Nanostructures – Fundamentals and Applications, 30 April 2018

3. "A visible light RGB wavelength demultiplex based on silicon-nitride multicore PCF",Optics & Laser Technology, 111, 2019 (411-416)

Author Response

1. The information for the GaN LED structure is not enough. The authors should give much more information about this which means that authors need to add the full LED structure, so the readers can get its reproducibility. 

R1: We have added the schematics of InGaN/GaN LED with LPD-Ag NP/ZnO window layer (Fig. 1) in page 3.

2. The authors should give much more information about the novelty of this paper, especially the effect of using these materials as LED, which applications can be used this device?

R2: High temperature process is a well-known method to transform the Ag thin film to Ag NPs. However, It also causes redistribution of dopant and introduces thermal stress to degrade the performance of device. We are the first time to synthesize the Ag NP at room temperature. Therefore, we have re-written the paragraph as "A well-know method to obtain Ag NPs is the solid-state dewetting process, which transform an Ag thin film into Ag NPs under a high annealing temperature. However, high-temperature process may result redistribution of dopant and thermal stress, which degrade the performance of devices. We are the first time to synthesize a liquid-phase-deposited Ag nanoparticle embedded zinc oxide (LPD-Ag NP/ZnO) thin film through chemical reaction in an aqueous solution at a low deposition temperature (i.e., even at room temperature)" in line 82-87 in page 2.

3. The fabrication tolerance analysis, which can offer a good guide for the fabrication requirement, and the key parameters, need to be added in the results section.

R3: The pH scale, HCl concentration and accuracy of deposition temperature are the important parameters to fabricate LPD-ZnO and LPD-Ag NP/ZnO. We have added them in line 132-135 in page 4.

4. More references need to be included in the introduction part to understand the applications of using GaN materials as a waveguide and LED as a light source in optical system.

     R4: We have added reference 5-7 and discussion in line 36-40 in page 1.

Round 2

Reviewer 1 Report

Authors made all requested changes. Therefore, I recommend this paper for publishing in Micromachines in the current form.

Reviewer 2 Report

The submitted manuscript is well revised according to the reviewer's comment. Therefore, the revised manuscript is accepted as is.

Reviewer 3 Report

The modified version cam be published 

This manuscript is a resubmission of an earlier submission. The following is a list of the peer review reports and author responses from that submission.

Round 1

Reviewer 1 Report

Manuscript reports on fabrication and testing LPD-Ag/NP-ZnO modified InGaN/GaN LEDs for improving light extraction efficiency.

The presented approach is logical, up to some extend novel, and concludes with results which might be of interest to the community.

Unfortunately, manuscript requires revisions before making final recommendation.

In general, authors used outdated references not reflecting on the “rapid development” of this science and engineering field. There is only a few reference from between 2015-2018. Please update.

I recommend that authors change the order of Section 2.1, shall be Section 2.2, it will be more appropriate to consider first LED fabrication then its modification.

Fog.1 – details in inserts are difficult to read, shall be improved. Panels (b) and (d) are not mentioned in caption, relevant text shall be revised if needed.

Line 193 – I think that there shall be Eq.6 not Eq.5

Line 211, there is …Ag NPS is 80 nm, later on line 286, the same is mentioned to be 76.8 nm, please correct

Authors shall present the Ag NPs size distribution diagram.

Fig.3 – It is seen that calculated optical intensity for InGaN/GaN LEDs with the flake-shaped ZnO window layer is comparable to InGaN/GaN LEDs with Ag NPs (~80nm) distributed on the flake-shaped ZnO window layer. This fact/result was not critically discussed and analyzed considering all pros and cons for both approaches in manuscript. Why to choose Ag NP/ZnO window approach if the final outcome is not much improved as compared to ZnO flake window case? Authors shall explain in greater details the observed fluctuations in optical intensity change as Ag NP diameter. What are the numbers here? How big is the predicted intensity change percentwise? And how the NP size distribution will affect the results shown in Fig.3. BTW, was NP size distribution included in modeling? If not how the realistic NP size distribution will affect the expected optical intensity outcome when considering LSP resonance? Authors shall critically comment on the results for NP with diameter close to 80nm as there is significant intensity variation observed. Simply, by changing NP size a bit there is complete LSP effect cancelation. How authors envision that this drawback can be resolved in practice?           

I assume, that the InGaN/GaN LEDs with Ag NPs/ZnO window layer was tested not encapsulated. Authors shall critically evaluate how the required device encapsulation will affect the overall final device efficiency. The concern here is on possible surface passivation effect of NPs/ZnO layer by encapsulating material and how it can affect the LSP.

Manuscript requires proofreading to eliminate minor English grammar and typos, e.g. see line 55, there is ‘corn”, shall be “cone” and other places.

Author Response

Professor Dr. Po-Hsun Lei,

Institute of Electro-Optical and material Science,

National Formosa University, No. 64, Wunhua Rd., Huwei, Yunlin County 632

Taiwan

Nov. 15, 2018

Prof. Dr. Laura Lao,

Editor, Applied Sciences

Dear editor:

Enclosed please find the revised paper (applsci-385514-Revision) entitled " Using silver nanoparticle/ZnO thin films prepared by liquid phase deposition method to improve light extraction efficiency of GaN/InGaN light-emitting diodes " by Po-Hsun Lei, Chyi-Da Yang, Po-Chun Huang and Sheng-Jhan Yeh. The changes are made according to the referees’ comments as attached. Thanks.

Respectfully,

Po-Hsun Lei,

Institute of Electro-Optical and material Science,

National Formosa University, No. 64, Wunhua Rd., Huwei, Yunlin County 632

Taiwan, Republic ofChina

PHONE: +886-5-6315668

FAX: +886-5-6315604

Reply to reviewers’ comments (applsci-385514-Revision)

In the comments made by the reviewers, some technical comments made from the reviewers for this paper are useful. We thank the constructive comments and made the following revisions. The specific comments made by the reviewers and replies are listed as following. Besides, the modified sentences and words were marked in red color.

Reviewer 1's comments

1. In general, authors used outdated references not reflecting on the “rapid development” of this science and engineering field. There is only a few reference from between 2015-2018. Please update.

R1: We have re-written some reference reported from 2015 to 2018 in reference part red color.

2. I recommend that authors change the order of Section 2.1, shall be Section 2.2, it will be more appropriate to consider first LED fabrication then its modification.

R2: We thank reviwer`s comment, and we have changed the order of section 2.1 and 2.2 in page 2, line 89-93 and page 3, line 94-119  in red color.

3. Fig.1 – details in inserts are difficult to read, shall be improved. Panels (b) and (d) are not mentioned in caption, relevant text shall be revised if needed.

R3: We have re-written the description of Fig. 1 in page 4, line 148-151 in red color.

4. Line 193 – I think that there shall be Eq.6 not Eq.5.

R4: It is a typo, and we have changed it in page 6, line 201 in red color.

5. Line 211, there is …Ag NPS is 80 nm, later on line 286, the same is mentioned to be 76.8 nm, please correct.

R5: It is a typo. The average diameter of Ag NP is 76.8 nm and we have changed it in red color in page 9, line 298.

6. Authors shall present the Ag NPs size distribution diagram.

R6: We have added the Ag NP size distribution diagrams in the insets of Fig. 1 (c) and (d) in page 4.

7. Fig.3 – It is seen that calculated optical intensity for InGaN/GaN LEDs with the flake-shaped ZnO window layer is comparable to InGaN/GaN LEDs with Ag NPs (~80nm) distributed on the flake-shaped ZnO window layer. This fact/result was not critically discussed and analyzed considering all pros and cons for both approaches in manuscript. Why to choose Ag NP/ZnO window approach if the final outcome is not much improved as compared to ZnO flake window case? Authors shall explain in greater details the observed fluctuations in optical intensity change as Ag NP diameter. What are the numbers here? How big is the predicted intensity change percentwise? And how the NP size distribution will affect the results shown in Fig.3. BTW, was NP size distribution included in modeling? If not how the realistic NP size distribution will affect the expected optical intensity outcome when considering LSP resonance? Authors shall critically comment on the results for NP with diameter close to 80nm as there is significant intensity variation observed. Simply, by changing NP size a bit there is complete LSP effect cancelation. How authors envision that this drawback can be resolved in practice?  

R7: We have changed Fig. 3 as Fig. 3 (a), (b) and (c). Fig. 3 (a) exhibits the calculated intensity of InGaN/GaN LED with different Ag NP size of the Ag NP/ZnO window layer. Figs. 3 (b) and (c) show the distribution of electric field on textured-ZnO and Ag NP/ZnO window layer. The distribution of electric field for InGaN/GaN LED with Ag NP/ZnO window layer show an increasing electric field intensity as compared to that with textured-ZnO window layer.

8. I assume, that the InGaN/GaN LEDs with Ag NPs/ZnO window layer was tested not encapsulated. Authors shall critically evaluate how the required device encapsulation will affect the overall final device efficiency. The concern here is on possible surface passivation effect of NPs/ZnO layer by encapsulating material and how it can affect the LSP.

R8: In this manuscript, we want to show a novel, simple and low-temperature method to improve the external quantum efficiency of InGaN/GaN LED. The discussion of encapsulation material and package technology, such as semi-sphere or parabolic shape, is not the main subject in this manuscript. However, we think that it can be a future work to investigate and discuss the effect of encapsulation material and related package technology.

9. Manuscript requires proofreading to eliminate minor English grammar and typos, e.g. see line 55, there is ‘corn”, shall be “cone” and other places.

R9: It is a typo, and we have change it as "cone" in page 2, line 54.

Reviewer 2 Report

The authors present a report on increasing the light extraction efficiency of InGaN/GaN MQW LEDs using a combination of surface roughening and localised surface plasmons of silver nano particles.

I have a few comments on the results of the paper that I would like clarified.

Firstly the authors describe how changing the PH of solution changes the morphology of the deposited ZnO flakes deposited on the surface of the LED. Has the effect of this change on the surface roughening on the light extraction efficiency been studied separately to the inclusion of the Ag nano particles?

In addition the stated variation in between LED chips is stated to be ~5%, however this is very close to the observed variation in EL intensity shown in figure 6. As such it is very difficult to de couple the effects of the ZnO flakes surface roughening and the Silver nano particles. It is my opinion that you should compare only against the 0M AgNO3 sample when comparing the final light output rather than the un-roughened device, and as such the figure of a 52% increase in total light emission is misleading , as only at most a few % of this is due to the Ag nano particles, with the remainder due to the effects of surface roughening which is a widely used and studied technique already.

In your introduction your summary of the improvements in InGaN GaN devices is rather poorly referenced, especially in the context of surface roughening techniques which are most relevant for comparison with this paper.

On Page 2 line 54 you refer to a “light escape corn” this needs to be corrected to “light escape cone”.

Figure 3 needs to include some numbers on the axis, or at least a 0 point. The figure as presented could represent either a 10 fold increase in device intensity, or a variation within the noise of measured devices, without some scale it this figure does not convey any useful information.

The marks denoting which axis the lines are plotted against in figure 4 are ambiguously placed.

Again Figure 6 needs to have some scale numbers included. Also this figure is in a different format to the others included in this paper.

Author Response

Professor Dr. Po-Hsun Lei,

Institute of Electro-Optical and material Science,

National Formosa University, No. 64, Wunhua Rd., Huwei, Yunlin County 632

Taiwan

Nov. 15, 2018

Prof. Dr. Laura Lao,

Editor, Applied Sciences

Dear editor:

Enclosed please find the revised paper (applsci-385514-Revision) entitled " Using silver nanoparticle/ZnO thin films prepared by liquid phase deposition method to improve light extraction efficiency of GaN/InGaN light-emitting diodes " by Po-Hsun Lei, Chyi-Da Yang, Po-Chun Huang and Sheng-Jhan Yeh. The changes are made according to the referees’ comments as attached. Thanks.

Respectfully,

Po-Hsun Lei,

Institute of Electro-Optical and material Science,

National Formosa University, No. 64, Wunhua Rd., Huwei, Yunlin County 632

Taiwan, Republic ofChina

PHONE: +886-5-6315668

FAX: +886-5-6315604

Reply to reviewers’ comments (applsci-385514-Revision)

In the comments made by the reviewers, some technical comments made from the reviewers for this paper are useful. We thank the constructive comments and made the following revisions. The specific comments made by the reviewers and replies are listed as following. Besides, the modified sentences and words were marked in red color.

Reviewer 2's comments

1. Firstly the authors describe how changing the PH of solution changes the morphology of the deposited ZnO flakes deposited on the surface of the LED. Has the effect of this change on the surface roughening on the light extraction efficiency been studied separately to the inclusion of the Ag nano particles?

R1: The roughness of LPD-ZnO thin film depends on the pH scale of the starting treatment solution. Higher or lower pH scale will affect the roughness of LPD-ZnO thin film that degrade the performance of InGaN/GaN LED in our previous study. To obtain a reliable LPD-ZnO thin film, the pH scale of the starting treatment solution should maintain fairly at 4-5.5. We have re-written this description in page 4, line 140-146.

2. In addition the stated variation in between LED chips is stated to be ~5%, however this is very close to the observed variation in EL intensity shown in figure 6. As such it is very difficult to de couple the effects of the ZnO flakes surface roughening and the Silver nano particles. It is my opinion that you should compare only against the 0M AgNO3 sample when comparing the final light output rather than the un-roughened device, and as such the figure of a 52% increase in total light emission is misleading , as only at most a few % of this is due to the Ag nano particles, with the remainder due to the effects of surface roughening which is a widely used and studied technique already.

R2: We thank reviewer`s suggestion. We have re-draw Fig. 5 and shown the standard deviation of measured enhancement of emission intensity of InGaN/GaN wafer with LPD-Ag NP/ZnO to that with LPD-ZnO.

3. In your introduction your summary of the improvements in InGaN GaN devices is rather poorly referenced, especially in the context of surface roughening techniques which are most relevant for comparison with this paper.

R3: We have added reference 15-18 and 28 in page 2, line 52 and 73.

4. On Page 2 line 54 you refer to a “light escape corn” this needs to be corrected to “light escape cone”.

R4: It is a typo, and we have changed it as "cone" in page 2, line 54.

5. Figure 3 needs to include some numbers on the axis, or at least a 0 point. The figure as presented could represent either a 10 fold increase in device intensity, or a variation within the noise of measured devices, without some scale it this figure does not convey any useful information.

R5: We have re-draw Fig. 3 and added the number in y-axis in Fig. 3 (a) in page 6.

6. The marks denoting which axis the lines are plotted against in figure 4 are ambiguously placed.

R6: We have re-draw Fig. 4 in page 7.

7. Again Figure 6 needs to have some scale numbers included. Also this figure is in a different format to the others included in this paper.

R7: We have added numbers in the intensity axis in Fig. 6 in page 8.

Reviewer 3 Report

Comments:

The authors reported on how to improve the performance of InGaN/GaN light-emitting diodes using liquid-phase-deposited Ag nanoparticles/ZnO. However, the data do not explain why Ag nanoparticles should be present on ZnO film. The authors attempted to explain it through the localized surface plasmon, but Ag nanoparticles of 50 nm or less that maximize the LSP were not synthesized. Also, Ag nanoparticles (unstable and easily agglomerated because of high surface energy) were not uniformly dispersed on ZnO flake by SEM analysis. The ZnO flake with Ag nanoparticles do not show sufficient improvements in LED performance. The manuscript does not have enough data to solve the reader’s curiosity. So, I though this manuscript was not suitable for Applied Sciences at current form.

Author Response

Professor Dr. Po-Hsun Lei,

Institute of Electro-Optical and material Science,

National Formosa University, No. 64, Wunhua Rd., Huwei, Yunlin County 632

Taiwan

Nov. 15, 2018

Prof. Dr. Laura Lao,

Editor, Applied Sciences

Dear editor:

Enclosed please find the revised paper (applsci-385514-Revision) entitled " Using silver nanoparticle/ZnO thin films prepared by liquid phase deposition method to improve light extraction efficiency of GaN/InGaN light-emitting diodes " by Po-Hsun Lei, Chyi-Da Yang, Po-Chun Huang and Sheng-Jhan Yeh. The changes are made according to the referees’ comments as attached. Thanks.

Respectfully,

Po-Hsun Lei,

Institute of Electro-Optical and material Science,

National Formosa University, No. 64, Wunhua Rd., Huwei, Yunlin County 632

Taiwan, Republic ofChina

PHONE: +886-5-6315668

FAX: +886-5-6315604

Reply to reviewers’ comments (applsci-385514-Revision)

In the comments made by the reviewers, some technical comments made from the reviewers for this paper are useful. We thank the constructive comments and made the following revisions. The specific comments made by the reviewers and replies are listed as following. Besides, the modified sentences and words were marked in red color.

Reviewer 3's comments

1. The authors reported on how to improve the performance of InGaN/GaN light-emitting diodes using liquid-phase-deposited Ag nanoparticles/ZnO. However, the data do not explain why Ag nanoparticles should be present on ZnO film. The authors attempted to explain it through the localized surface plasmon, but Ag nanoparticles of 50 nm or less that maximize the LSP were not synthesized. Also, Ag nanoparticles (unstable and easily agglomerated because of high surface energy) were not uniformly dispersed on ZnO flake by SEM analysis. The ZnO flake with Ag nanoparticles do not show sufficient improvements in LED performance. The manuscript does not have enough data to solve the reader’s curiosity. So, I though this manuscript was not suitable for Applied Sciences at current form.

R1: We have re-written the manuscript to make it be suitable for Applied Sciences. In addition, the size of Ag NPs in our manuscript is larger than 50 nm.

Round 2

Reviewer 1 Report

This is revised manuscript. Authors attempted to address all reviewer’s concerns, thank you.

Q:7 was not addressed satisfactorily yet, specifically authors have not commented on: “…the results for NP with diameter close to 80nm as there is significant intensity variation observed”. Furthermore, Fig.3 has been revised by adding the electric field distributions for tested LEDs but the meaning of this new result has not be discussed beyond introducing it in the text; thus its significance is not obvious to the reader. Closer look at Fig.3(a) reviled that the authors have modified this plot artificially, at least it appears to look like. The guide to the eyes (line connecting data points) is now misleading. The data point at ~70nm is before ~74nm then at ~80nm. Thus, the Fig.3(a) is different from its original version. Why? Therefore, reviewer holds his original Q7 concerns and expect that authors will address them critically to the fullest possible extent because the experiment demonstrated that the intensity fluctuation in case of NPs with the 80nm diam. fluctuates considerably.

Q8 still needs to be addressed/mentioned in manuscript, even it was not the main research objective of this report, especially considering very strong sensitivity of the observed LSP effect on NPs size distribution addressed in Q7. The main concern here is, will encapsulation affect the expected LSP effect or not? Authors shall comment on this when envisioning the future research even without experimental data available at this time.

BTW, the details of inserts in Fig.1(a),(c) and Fig.2 are not readable. Please correct.

I do not recommend manuscript for publication without correcting the above issue.

Reviewer 2 Report

I believe you have missed the point of my first question. In Figure 1 you show SEM images of your ZnO for several of your AgNO3 concentrations. The extent of the ZnO flakes varies drastically in both size and structure between these micrographs. This as such will vary the surface roughening of the sample, and as such the light extraction efficiency of these samples. In order to separate the effect of LSP from the surface roughening in this paper the effect of the crystal morphology needs to also be included in the paper to show that the observed variation in light extraction efficiency is not simply a factor of the changes in surface roughening.  Your additional comment does not currently alleviate this concern.

For my second comment, while the modification to figure 5 is appreciated, it does not address my concern that comparing your samples to the roughened samples is misleading. This is done throughout the paper and needs to be corrected. This would reduce your device improvement drastically from the 52% increase you currently claim.

Reviewer 3 Report

The author prepared the revised manuscript well and it is more clear now for readers. So this manuscript can be published in Applied Science as current form.